# The Relationships between Self-Reported Motor Functioning and Autistic Traits: The Italian Version of the Adult Developmental Coordination Disorders/Dyspraxia Checklist (ADC)

**DOI:** 10.3390/ijerph20021101

**Published:** 2023-01-08

**Authors:** Isa Zappullo, Massimiliano Conson, Chiara Baiano, Roberta Cecere, Gennaro Raimo, Amanda Kirby

**Affiliations:** 1Department of Psychology, University of Campania Luigi Vanvitelli, 81100 Caserta, Italy; 2IRCCS San Camillo Hospital, 30126 Venice, Italy; 3Do-IT Solutions, London WC2H 9JQ, UK

**Keywords:** developmental coordination disorders, dyspraxia, autistic traits, systemizing, empathy

## Abstract

Background: We developed the Italian version of the adult developmental co-ordination disorders/dyspraxia checklist (ADC), providing reliability and concurrent validity data for the scale (Aim 1). In addition, we investigated the relationships between motor coordination difficulties and different autistic traits (Aim 2). Methods: 498 participants (341 females; age range = 18–34) underwent the Italian version of the ADC, as well as a handwriting speed test, the autism spectrum quotient, the empathy quotient, and the systemizing quotient. Results: The distinction between three main factors (A, B, and C) identified in the original version of the ADC was confirmed here. Internal consistency of the ADC subscales was adequate, as well as the correlation with the handwriting speed test used to assess concurrent validity. No substantial sex differences on the ADC scores were found. Furthermore, data revealed that poor autistic-related communication skills and lower levels of systemizing tendencies were, among all the investigated autistic traits, those more strongly associated with higher motor coordination difficulties. Conclusions: The Italian ADC seems a valuable tool for assessing motor coordination difficulties in adults and can reveal the nuanced impact exerted by different autistic traits on self-reported motor functioning.

## 1. Introduction

Developmental coordination disorder (DCD), also known as dyspraxia, is a neurodevelopmental disorder characterized by difficulties in several aspects of motor coordination, as selection, timing and spatial organization of purposeful movements and coordination [1].

According to the *Diagnostic and Statistical Manual of Mental Disorders* (DSM-5, 5th ed.) [2] classification for DCD, dyspraxic children have fine and/or gross motor skills below the level expected for their age and learning opportunities; these motor impairments are not better explained by any other medical, neurodevelopmental, psychological, social condition, or cultural background, occur in the early developmental period and interfere with several areas of daily living, such as school, work productivity, home life, play, and leisure activities. 

Available research on DCD has largely focused on school-age children, demonstrating the impact of DCD spreading beyond the motor domain since individuals with DCD can present higher rates of psychological, social, educational, and health problems with respect to their typically developing peers (for a review see [1]). However, DCD is a lifelong condition, with most of the individuals showing significant motor difficulties through adolescence and into adulthood negatively impacting on management of daily living activities and on health-related quality of life [3,4,5,6]. For example, many higher-education students with DCD still report relevant problems in handwriting and organizational abilities [7]. In addition, young adults with DCD report socialization problems to a greater extent than their typical peers [8], which may increase the risk of developing clinical anxiety and depression [9]. Furthermore, challenges related to education or employment may result in psychological and social issues undermining young adults’ mental health, life satisfaction, and self-esteem [6,10]. The impact of DCD on the individual’s functioning can be strongly increased by the co-occurrence with other developmental disorders [9]. Comorbidity in DCD is frequently observed with attention-deficit hyperactivity disorder (ADHD) and autism spectrum conditions [11]. In adults, recent evidence from a large population sample of adults revealed increased self-reported diagnosis of dyspraxia in individuals with clinical autism and in neurotypicals with higher autistic traits [12].

One of the few existing self-report measures for DCD in adults is the adult developmental co-ordination disorders/dyspraxia checklist (ADC), developed by Kirby and colleagues [13]. The ADC is a valuable tool for diagnostic, treatment, and research issues in adults with DCD, and, at the present, is available in English, Hebrew [13], and German [14]. It is important to make the ADC available for further cultural groups and in other languages [13,14]. In Italian, self-report measures for DCD only exist for children [15,16] and, to our knowledge, no self-report measures target the adult population. Therefore, one purpose of the present study was the translation and assessment of ADC to make this scale available to Italian-speaking adults for the first time. The second aim was to assess, in the general population, the impact of different kinds of autistic traits on DCD-related symptoms. In detail, the following two main aims were targeted.

*Aim 1*. Developing the Italian version of the ADC for a population of individuals aged between 18 and 34 years, providing reliability and concurrent validity data for the scale. Both age and sex differences have been found in adults screened for probable and at-risk DCD using the ADC [17]. Thus, here, sex differences in the Italian ADC scores were considered, while no age effects were assessed since the age range of our population was narrow. The original version of the ADC [13] has an important focus on handwriting as a main everyday living activity requiring good motor coordination, and in the original study on ADC concurrent validity of the scale was tested by comparing the ADC with the score on a handwriting self-report questionnaire (the handwriting proficiency screening questionnaire) [18] assessing the most important indicators of handwriting difficulties. Problems with handwriting are among the criteria for the diagnosis of DCD according to the DSM-V [2]. Consistently, handwriting difficulties in school-aged children contribute to limitations in their participation in the academic context [19,20]. On this basis, in the present study concurrent validity of the Italian ADC was determined by comparing the scores of the scale with the score on a handwriting task widely used in Italy to assess handwriting efficiency, i.e., the “writing numbers in letters” test (MT-16-19 Battery) [21].

*Aim 2*. Investigating the relationships between different autistic traits and DCD-related signs assessed by the Italian ADC. Available studies on the relationships between autistic traits and DCD symptoms in the general population are scarce. In a large sample study, Cassidy and colleagues [12] evaluated autistic traits by means of the autism spectrum quotient (AQ) [22], and the empathy quotient (EQ) [23]. More recently, Tal-Saban and Kirby [5], specifically assessed empathic skills through the EQ [23] in adults with DCD considering the comorbidity with ADHD and autism. Following these studies, here, the AQ and the EQ were used to assess autistic traits, but one more trait relating to the autism spectrum was also assessed, i.e., systemizing, by means of the systemizing quotient (SQ) [24]. Systemizing is defined as the tendency to analyze, comprehend, and build systems through the implementation of ‘if-and-then’ logical reasoning, a strength in people on the autism spectrum (hyper-systemizing) [25,26]. Due to the relevance of systemizing in characterizing the cognitive style of individuals on the autism spectrum [24], in the present study it was explored whether and how systemizing can relate to DCD signs assessed by the Italian version of the ADC.

## 2. Materials and Methods

### 2.1. Participants

A sample of 300 subjects is generally sufficient to ensure adequate statistical power for confirmatory factor analysis (CFA), but the ratio of 10 cases per observed variable has also been considered [27,28,29]. Therefore, to be sure that the sample size of 400 subjects would be sufficient for subsequent statistical techniques (Pearson correlation, MANOVA, and multiple regression), separate a priori power analyses were carried out (g* power) [30] to guarantee detection of an effect of critical interest with a power of 0.80 at an alpha level of 0.05 with a medium effect size (0.30, 0.25, and 0.15, respectively). Results indicated that 82 subjects were needed for both the Pearson correlation and the MANOVA, whereas a sample of 103 subjects were required to detect an effect of critical interest in the regression analysis. The sample recruited for the study included 498 Italian language-speaking participants (341 females, 157 males; mean age = 23.43; SD = 2.96; range = 18–34) and all of them met the following inclusion criteria: (i) lack of a past diagnosis of neurological or neurodevelopmental conditions, and (ii) lack of any history of psychiatric difficulties, defined on the basis of participants’ self-reports. Each participant was individually assessed in a quiet room over a single session lasting about 45 min (raw data are available upon request from the first author). The study was conducted in accordance with the standards of the Helsinki declaration and approved by the local Ethics Committee of the Department of Psychology, University of Campania Luigi Vanvitelli (protocol code: N:30/2020). Participants’ written informed consent was obtained before starting the study.

### 2.2. Materials

*Adult Developmental Co-ordination Disorders/Dyspraxia Checklist (ADC)*. The ADC [13] is a 40-item self-report measure, evaluating the main areas of performance in which adults with developmental co-ordination disorder (DCD) characteristically experience difficulties, including activities of daily living, handwriting, driving, attention, organizing activities in time and space, and social skills. The ADC provides 3 factors and a total score. The first subscale (ADC-A) is composed by 10 items measuring difficulties that individuals experienced as a child. The second subscale (ADC-B) includes 10 items, and the third subscale (ADC-C) is composed by 20 items, both relating to current difficulties experienced by the individual. The difference between subscales B and C lies in that subscale B focuses on the influence of motor coordination abilities on the individual’s perception of her/his performance, while subscale C more specifically includes social consequences of symptoms; as the authors describe it, symptoms “manifested by others” [13]. Each item describes a difficulty, and the participant is asked to indicate on a 4-point Likert scale whether this difficulty occurs “never” (1), “sometimes’ (2), “frequently” (3), or “always” (4), thus higher scores indicate greater motor coordination difficulties. Cronbach’s α coefficients showed a good internal reliability (ADC-A = 0.91; ADC-B = 0.87; ADC-C = 0.90) [13]. 

The original English-language 40-item ADC [13] was translated into Italian by two independent Italian researchers who adapted the items to the Italian cultural contest. The two translations were then reviewed to fix discrepancies, resulting in a single agreed-upon Italian version. This translation was then back translated into English by a bilingual researcher blinded to the original English version. The two versions (English and Italian) were carefully compared, and the Italian one was reviewed by an expert committee, correcting items that were unclear. Finally, the resulting Italian version was tested on a sample of 20 participants to verify the clarity and face validity of the items (Appendix A).

*Handwriting*. A task widely used in Italy to assess handwriting speed, “writing numbers in letters” (MT-16-19 Battery) [21], was selected here to evaluate handwriting skills. Participants are required to write the numbers in letters as quickly as possible in ascending order within 60-second time limit. The total number of graphemes correctly written within the time limit is recorded. The “writing numbers in letters” task showed a good test-retest reliability (r = 0.770) [21].

*Autism Spectrum Quotient (AQ)*. The AQ [22,31] measures the number of autistic traits an individual possesses across five domains (social skill, attention switching, attention to detail, communication, and imagination) in both clinical and non-clinical samples. Participants were administered the full 50-item AQ [22]. The results were scored according to Baron-Cohen et al.’s [22] criteria, resulting in a total AQ score and in further five scores for the corresponding five subscales (social skill, attention switching, attention-to-detail, communication, and imagination), with higher scores indicating poor social skill, poor attention-switching/strong focus of attention, exceptional attention to detail, poor communication, and poor imagination skills, respectively. Cronbach’s α coefficients demonstrated an acceptable internal consistency (AQ total = 0.76, social skills = 0.68, attention switching = 0.54, attention-to-detail = 0.58, communication = 0.64, and imagination = 0.52) and good test–retest reliability at 6 months (Pearson r = 0.79) [31].

*Empathy Quotient (EQ)*. The EQ [23,32] evaluates empathy traits related to the recognition of others’ emotions and moods. Participants answered the 40-item short version of the EQ. The results were scored to obtain a total EQ score which represents the subjects’ level of empathy traits, with high scores indicating greater ability to understand others’ emotions and moods. The Italian version of the EQ showed good reliability (Cronbach’s α = 0.79; test–retest at 1 month: Pearson r = 0.85) [32].

*Systemizing Quotient (SQ).* The SQ [24] measures across separate examples of systemizing the individual’s interest in a range of systems. The SQ comprises 60 questions: 40 assessing systemizing and 20 filler (control) items. Here, the Italian translation of the scale was used, which is published on the website of the Cambridge Autism Research Centre (https://www.autismresearchcentre.com/ (accessed on 1 November 2021)). The results provide a total SQ score indicating individual differences across the systemizing dimension, with high scores indicating greater systemizing. Cronbach’s α coefficient of the SQ was 0.79 [24].

### 2.3. Statistical Analysis

#### 2.3.1. Aim 1: Psychometric Properties of the Italian ADC

Confirmatory factor analysis (CFA) was carried out to determine whether the theoretical three-dimensional structure of the ADC fitted the observed data. The items from A1 to A10 were specified to load on the first factor (ADC-A), items from B1 to B10 were specified to load on the second factor (ADC-B) whereas items from C1 to C20 were specified to load on the third factor (ADC-C). To obtain robust parameter estimates, asymptotic covariance matrices and the maximum likelihood method were used. To define potentially significant parameters to add, modification indexes (MIs) [33] of the tested model were also considered. The model including all the relevant parameters was considered as the reference model. As for fit indices, the maximum likelihood (MLχ^2^) goodness-of-fit test statistics in combination with the root mean square error of approximation index (RMSEA) and the comparative fit index (CFI) [33,34] were used. The following values were considered to indicate acceptable fit: values < 0.08 for RMSEA; values > 0.90 for CFI [33,35]. CFA was performed with LISREL 8.71 software [36]. 

Reliability of the Italian version of the ADC subscales were evaluated using Cronbach’s alpha. To evaluate concurrent validity of the scale, the handwriting speed test was administered to a subsample of 260 participants (171 females, 89 males; mean age = 23.60; SD = 2.84; range = 18–34; ADC-A: M = 16.09, SD = 4.57; ADC-B: M = 15.83, SD = 4.41; ADC-C: M = 36.13, SD = 7.14; ADC-Total score: M = 68.06, SD = 13.59). Therefore, Pearson correlation coefficients between the ADC total and subscales scores and the handwriting speed test were executed. Cohen’s guidelines [37] were considered for interpreting the magnitude of a correlations (r = 0.10, r = 0.30, and r = 0.50 were small, medium, and large in magnitude, respectively). Possible sex differences on the ADC subscales were assessed with a MANOVA conducted on the ADC total score and on the three ADC subscales, with sex (females vs. males) as the between-subjects factor. These analyses were performed by the Statistical Package for Social Sciences (SPSS Inc, version 22.0, Chicago, IL, USA).

#### 2.3.2. Aim 2: Relationships between ADC and Autistic Traits

Preliminary descriptive analyses were carried out to examine missing values and variables distributions. Pearson correlation coefficients between the ADC total and subscales scores (ADC-A, ADC-B, ADC-C, and ADC-Total score) and the autistic traits measures (AQ-social skill, AQ attention switching, AQ attention-to-detail, AQ communication, and AQ imagination subscales, EQ, and SQ scores) were executed to investigate bivariate relations between variables. Then, regression analyses were performed to test which autistic traits could predict ADC scores.

Given the correlation between the predictors, here was adopted the least absolute shrinkage and selection operator (LASSO) regression model [38], a regression technique in which the coefficients are biased from a penalty term to enhance the prediction accuracy and interpretability of the statistical model. The LASSO is a linear model with regularization to reduce model complexity and avoid over-fitting in prediction model. In the LASSO, parameters are shrunk toward a central point and this applied penalty increases the efficiency in variable selection and parameter elimination, preserving only the most relevant coefficients. Indeed, through this technique some coefficients can become zero and are eliminated from the model. Given the specificity of the LASSO method, which does not provide tests of significance, the non-zero coefficients can be interpreted as the importance of the variable in contributing to the underlying variation of the data: the higher the absolute value of a coefficient, the more important the weight of the corresponding variable [38,39,40]. Four separate LASSO models were executed, with autistic traits as independent variables and with the ADC scores one by one as dependent variables. Prior to analysis, data across all measures were normalized as z scores. The degree of shrinkage was determined by a 5-fold cross-validation, and a sub-sample of a 100 randomly selected observations was set to validate the model. As for fit indices, the determination coefficient (R2), the root mean square error index (RMSE) and the mean square error (MSE) were used [40]. These analyses were performed with XLSTAT package [40].

## 3. Results

### 3.1. Aim 1: Psychometric Properties of the Italian ADC

#### 3.1.1. Confirmatory Factor Analysis

Results of the first CFA did not show a good fit for the 40 items modelled in terms of three factors: MLχ^2^(740) = 3893.8; *p* < 0.001; RMSEA = 0.093; CFI = 0.80. The analysis of modification indices (MIs) indicated that ADC-A and ADC-C (Standardized ψ = 0.769; *p* < 0.001) and ADC-B and ADC-C (standardized ψ = 0.372; *p* < 0.001) factors were significantly correlated, as well as the error terms of some of the items: i.e., items A5 and C6 (standardized ε = 0.510; *p* < 0.001); items A9 and B8 (standardized ε = 0.380; *p* < 0.001); items A7 and C3 (standardized ε = 0.349; *p* < 0.001); items B1 and B2 (standardized ε = 0.317; *p* < 0.001); items B3 and C7 (standardized ε = 0.433; *p* < 0.001); items C5 and C12 (Standardized ε = 0.519; *p* < 0.001); items C5 and C18 (standardized ε = 0.290; *p* < 0.001); items C9 and C10 (standardized ε = 0.455; *p* < 0.001) and items C12 and C18 (standardized ε = 0.477; *p* < 0.001). Thus, these additional paths were included in the 40-item three-factor model that was considered as the new 40-item three-factor model and the fit of the model was tested again. Results of this CFA showed an adequate fit for the corrected model that considered all the significant paths between items, MLχ^2^(729) = 2353.40; *p* < 0.001; MLχ^2^/df = 3.22; RMSEA = 0.067, 90% CI [0.063; 0.070]; ECVI = 5.10; CFI = 0.90. The standardized item saturations ranged from 0.553 to 0.218 (M = 0.385) for the ADC-A subscale, from 0.501 to 0.232 (M = 0.394) for the ADC-B subscale and from 0.398 to 0.141 (M = 0.289) from the ADC-C subscale (Table 1).

#### 3.1.2. Reliability

The subscales showed an adequate internal consistency: Cronbach αs were 0.756, 95% CI [0.723; 0.787] for the ADC-A subscale, 0.790, 95% CI [0.761; 0.816] for the ADC-B subscale and 0.794, 95% CI [0.767; 0.819] for the ADC-C subscale.

#### 3.1.3. Concurrent Validity 

Pearson correlation coefficients indicated that handwriting speed (*N* = 260; M = 112.4, SD = 23.25; range: 36–185 graphemes) was negatively associated with the ADC-A, r = −0.208; *p* = 0.001; 90% CI [−0.306; −0.109] and the ADC-B, r = −0.156; *p* = 0.012; 90% CI [−0.256; −0.055] subscales and with the ADC-Total score, r = −0.128; *p* = 0.040; 90% CI [−0.228; −0.027]. No significant correlation was found between handwriting speed and the ADC-C subscale, r = −0.013; *p* = 0.835; 90% CI [−0.115; 0.089].

#### 3.1.4. Sex Differences

The mean scores for men and women on the ADC total score and subscales are reported in Table 2. Results of a MANOVA revealed a small effect of sex on ADC-C subscale, F(1, 496) = 4.09, *p* = 0.044, *η^2^_p_* = 0.008, whereas no significant effects of sex on either ADC-A, F(1, 496) = 1.31, *p* = 0.252, *η^2^_p_* = 0.003, and ADC-B, F(1, 496) = 0.13, *p* = 0.715, *η^2^_p_* = 0.000, subscales or ADC-Total score, F(1, 496) = 0.33, *p* = 0.561, *η^2^_p_* = 0.001.

### 3.2. Aim 2: Relationships between ADC and Autistic Traits

#### 3.2.1. Correlation Analysis

Descriptive analyses are reported in Table 3. Pearson correlation coefficients (Table 4) indicated that the ADC-A and ADC-B subscales and the ADC-Total score were positively correlated with social skill, attention switching, communication and imagination subscales of the AQ, and negatively related to both EQ and SQ, whereas the ADC-C subscale was positively related to social skill, attention switching and communication subscales of the AQ, and negatively correlated with both EQ and SQ.

#### 3.2.2. LASSO Regression Analysis

All the four LASSO models indicated an adequate fit (Table 5). The zero-correlation coefficients, eliminated from their model, were respectively: (i) AQ social skill for the first model (ADC-A); AQ attention-to-detail for the second (ADC-B) and third (ADC-C) model; AQ imagination for the third model (ADC-C). Although all the remaining coefficients contributed (with different degrees) to the underlying variation of the data, the variables that showed a higher weight in all models were AQ communication and SQ, indicating that both poor communication skills and lower levels of systemizing were more strongly associated with higher motor coordination difficulties (Figure 1).

## 4. Discussion

Aim 1 of the present study was to provide the psychometric properties of the Italian version of the ADC. First, CFA was performed to find out whether the theoretical three-dimensional structure of the original version of the ADC [13] fitted the observed data. Results demonstrated an adequate fit for the 40-item three-factor model, thus confirming the distinction between A, B and C factors of the original version of the ADC [13]. Recently, Meachon and colleagues [14] suggested that to assess individuals with DCD, manifested either in isolation or in comorbidity with ADHD, a restructuration of the ADC from its original three subscales to a new set of three subscales based on symptomatic aspects was needed. Therefore, the potential use of the Italian ADC to identify clinical DCD and distinguish cases co-occurrence of DCD with other neurodevelopmental conditions, such as ADHD and autism, should be directly tested in future research.

Assessment of reliability of the Italian version of the ADC subscales demonstrated an adequate internal consistency for each of the three subscales. Furthermore, evaluation of concurrent validity of the scale demonstrated that handwriting speed was negatively correlated with scores on the subscales A and B and with the total score of the ADC, whereas no significant relationship was found between handwriting speed and the ADC-C subscale. Kirby and colleagues [13] found that all the three ADC subscales were moderately correlated with the score on a self-report questionnaire assessing the most important signs of handwriting difficulties [18]. Here, handwriting proficiency was assessed by means of an objective measure of writing efficiency (“writing numbers in letters” test), focusing on handwriting speed (MT-16-19 Battery) [21]. This aspect might explain the difference between the present and Kirby and colleagues’ [13] results with respect to the lack of correlation between ADC-C and handwriting speed found here. Indeed, ADC-C specifically includes social consequences of symptoms (i.e., symptoms “manifested by others”) [13], while ADC-B evaluates the influence of motor coordination skills on the individual’s perception of her/his performance. Our data suggest that the subjective perception of handwriting difficulties measured by the ADC-B adequately reflects the true and objective impact of motor difficulties on handwriting speed that can be differentiated from the social perception of motor coordination-related difficulties in handwriting.

Finally, possible sex differences were assessed on the ADC total score and the three ADC subscales, and results reported no substantial sex differences on both the ADC subscales and the total score, with the only exception of a slight difference on the ADC-C subscale, with a higher score in women than men. Recently, Cleaton and colleagues [17] investigated sex differences in adults screened for probable and at-risk DCD using the ADC and found in participants who suspected they had DCD a significantly greater percent of women reporting higher current difficulties than men, notwithstanding a similar percentage of childhood difficulties. The authors only distinguished between childhood (ADC-A subscale) and current (ADC-B and C subscales collapsed together) difficulties; thus, it was not possible to establish whether B or C subscales could highlight differences relating to sex. The present results are generally consistent with Cleaton and colleagues’ [17] ones, and also allow to suggest that women on average could be more able than men to evaluate the social impact of their motor difficulties. Although the present research did not look at the specific motor abilities highlighting sex differences, the above interpretations fit with data suggesting that women with DCD might be more exposed than men to social disapproval due to clumsiness or to specific expectations greatly impacting on their social activities and participation [17].

Aim 2 of the study was to investigate the relationships between different autistic traits and DCD signs assessed by the Italian ADC. Results of correlation analysis showed small-to-medium positive correlations between ADC scores and all AQ subscales apart from AQ attention-to-detail which did not correlate with any of the ADC scores, whereas small-to-medium negative correlations were found between ADC scores and both SQ and EQ. More importantly, results of the regression analysis revealed that poor communication skills (higher AQ communication scores) and lower levels of systemizing (lower SQ scores) were, among all the investigated autistic traits, those more strongly associated with higher motor coordination difficulties (higher scores on all the ADC scales).

In a large sample study investigating relationships between autistic traits and motor coordination difficulties in adults with both clinical and non-clinical autism, Cassidy and colleagues [12] found that the frequency of self-reported dyspraxia was significantly higher in individuals with autism than in those without autism, whereas the two groups did not differ with respect to AQ or EQ scores. Relevantly here, the authors also found in the group without autism that individuals with dyspraxia had significantly higher AQ and lower EQ scores than those without dyspraxia. Our results are consistent with Cassidy and colleagues’ [12] findings on neurotypical adults who reported having received a diagnosis of dyspraxia. The neurotypical adults recruited for the present study did not report a previous diagnosis of neurodevelopmental disorders, but DCD signs were measured through the Italian ADC. Relevantly, moreover, whether specific autistic traits were more related to DCD signs was also assessed by exploiting the AQ subscales, together with empathy and systemizing questionnaires. No previous data is available on whether some specific autistic traits most relate to DCD symptoms, since, for instance, Cassidy and colleagues’ [12] only used the total score of AQ rather than its subscales. Our results showed that that the AQ subscale focusing on autistic-related communication difficulties was the most strongly associated with higher scores on all the ADC scale. However, positive medium correlations were also found between DCD scores and those from the AQ scales most pertaining the social dimension of the autistic traits (social skill, attention switching, communication, and imagination), while leaving uncorrelated the non-social dimension (attention-to-detail) of the traits [41]. An indirect support to our results only can be found in a recent study on interactional synchrony in neurotypicals with autistic traits. Granner-Shuman and colleagues [42] demonstrated in those with higher AQ communication scores interactional synchrony difficulties (assessed by a task requiring two participants to synchronize their hands movements) partially mediated by motor planning and execution abilities. 

The strong association between social autistic traits and motor functioning in neurotypicals are consistent with data on clinical autism highlighting a tight link between motor and social difficulties, although such a relationship has been clearly reported in studies investigating motor functioning through socially oriented motor tasks as praxis and imitation, while it appears more nuances when motor functioning is assessed through tasks focusing on gross and fine motor abilities [43,44,45,46].

Among the social traits, a negative relationship between EQ and ADC scores was found here, suggesting lower empathic abilities in those with stronger DCD-related difficulties. This finding was consistent with Cassidy and colleagues’ [12] data on neurotypical adults, whereas did not fit available data on adults with DCD [5,46]. Tal-Saban and Kirby [5] showed that adults with DCD did not differ on EQ from neurotypical controls whereas they differed from individuals with comorbid autism/DCD. More recently, Kilroy et al. [46] found that comorbid autism/DCD was specifically related to difficulties in cognitive rather than affective aspects of empathy. Cassidy and colleagues [12] suggested that a difference can exist between clinical and non-clinical populations in the relationship between motor coordination difficulties and empathy. This could be explained by a different strength in the association between these two aspects which would be more evident in people with non-clinical difficulties while being less evident in those with a co-morbid diagnosis of autism and DCD in which an overlap of symptoms of the two disorders could impede to observe difficulties of empathy and motor functioning separately [12]. Thus, our results can provide further support to this view. Importantly, moreover, the present findings prompt a specific discussion on the negative relationship between systemizing and motor coordination difficulties. 

Systemizing has been considered a strength in people on the autism spectrum (as posited by the hyper-systemizing model of autism) implying a strong drive to understand and predict the functioning of a system through the implementation of the “if-and-then” logical rule [25,26]. Independently from the specific nature of the system, the technical (as a computer), natural (as a weather front), abstract (as mathematics), or motoric (a sport technique or performance) functioning of a system can be operationalized into the logical “if-and-then” rule [24]. For instance, in the motoric domain, by implementing the “if-and-then” rule an individual can understand and execute the motor schemata underpinning complex movement patterns, such as those required in a sport [24,25]. Along the same line, it could be possible to extend the positive relationship between systemizing and motoric systems to several other complex motor behaviors, for instance as those required in writing, an ability assessed by the ADC and a weakness in people with DCD. In this framework, the strong negative relationship found here between SQ and ADC scores can be interpreted as reduced motor coordination difficulties in those with stronger systemizing tendences, consistently with predictions of the early conceptualization of systemizing [25].

### Limitations

There are two main limitations of the study which have to be considered. The first limitation lies in not having considered ADHD traits. Comorbidity with other developmental disorders is common in those with DCD. In particular, a strong comorbidity has been reported among DCD, ADHD and autism [5,47]. On the other hand, several features of DCD overlap with features of other neurodevelopmental disorders, especially ADHD (for a review see [48]). Recently, Meachon et al. [14] developed the German version of the ADC and evaluated its potential to distinguish DCD versus ADHD profiles. Results showed that the German version of the scale was useful in distinguishing DCD individuals from neurotypical adults. Furthermore, results showed a structure of the scale different from its original version that was effective in differentiating between adults with DCD and ADHD. The present findings cannot allow to establish the extent to which the Italian version of the ADC can differentiate between adults with DCD and ADHD. Thus, following Meachon et al.’s [14] findings, a specific study is needed to verify the potential for the Italian version of the ADC to deal with this important clinical application issue. 

The second limitation relates to the fact that participants could take part in the present study if they did not report any past diagnosis or history of neurological, neurodevelopmental, or psychiatric conditions. Since the study protocol did not include any diagnostic measure, no confirmatory diagnostic data could be collected to verify participants’ self-reports. Therefore, a further study should include an assessment process in order to better examine the diagnostic profiles of the participants. 

## 5. Conclusions

In the present study, the Italian version of the ADC for adults was developed. Results confirmed the distinction between three main factors (A, B, and C), consistent with the original structure of the scale [13]. Internal consistency of the three subscales was adequate, as well as the concurrent validity assessed by correlating the ADC with handwriting speed, while no considerable sex differences in the ADC scores were observed. The main results on the relationships between ADC and autistic traits showed that poor autistic-related communication skills and lower levels of systemizing tendencies were, among all the investigated autistic traits, those more strongly associated with higher motor coordination difficulties. The Italian ADC seems a valuable tool for evaluating motor coordination problems in adults, and it can be also useful to highlight the significant impact exerted by specific autistic traits on self-reported motor functioning.

## Figures and Tables

**Figure 1 ijerph-20-01101-f001:**
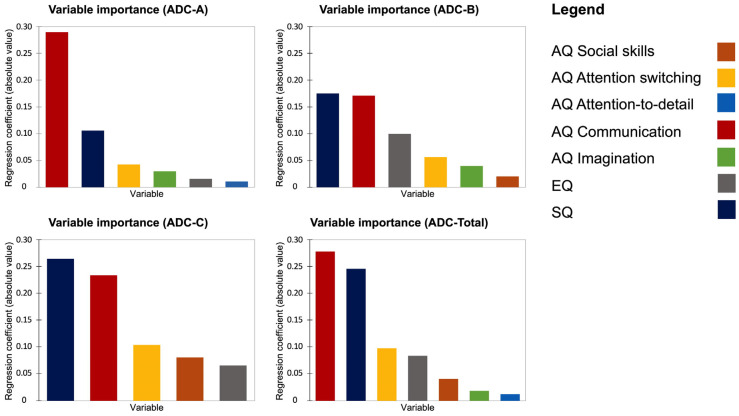
Charts of Variable importance for the four ADC LASSO regression models. The number of variables displayed on each model chart was calculated considering the number of non-zero coefficients in the model. Note that the importance measure for a given variable corresponds with the absolute value of its coefficient in the regression.

**Table 1 ijerph-20-01101-t001:** Standardized saturations of the Italian version of the ADC.

Factors			
Item	[1] ADC-A	[2] ADC-B	[3] ADC-C
A10	0.553 ***	-	-
A7	0.515 ***	-	-
A4	0.463 ***	-	-
A2	0.402 ***	-	-
A9	0.381 ***	-	-
A3	0.360 ***	-	-
A8	0.348 ***	-	-
A5	0.320 ***	-	-
A6	0.298 ***	-	-
A1	0.218 ***	-	-
B7	-	0.501 ***	-
B4	-	0.494 ***	-
B5	-	0.478 ***	-
B6	-	0.473 ***	-
B8	-	0.428 ***	-
B3	-	0.381 ***	-
B9	-	0.374 ***	-
B10	-	0.339 ***	-
B1	-	0.247 ***	-
B2	-	0.232 ***	-
C4	-	-	0.398 ***
C19	-	-	0.393 ***
C3	-	-	0.379 ***
C1	-	-	0.369 ***
C2	-	-	0.369 ***
C15	-	-	0.368 ***
C20	-	-	0.341 ***
C18	-	-	0.337 ***
C16	-	-	0.320 ***
C12	-	-	0.299 ***
C7	-	-	0.282 ***
C9	-	-	0.282 ***
C17	-	-	0.274 ***
C14	-	-	0.273 ***
C5	-	-	0.259 ***
C6	-	-	0.207 ***
C13	-	-	0.187 ***
C10	-	-	0.157 **
C8	-	-	0.143 **
C11	-	-	0.141 *

Note. * *p* < 0.05; ** *p* < 0.01; *** *p* < 0.001.

**Table 2 ijerph-20-01101-t002:** ADC total and subscale scores (mean and SD) as a function of sex.

Factors			
	Total Sample (*N* = 498)	Males (*N* = 157)	Females (*N* = 341)
ADC-A	16.16 (4.60) [9–33]	16.51 (5.31) [10–33]	16.00 (4.24) [9–32]
ADC-B	15.80 (4.52) [10–33]	15.90 (4.80) [10–32]	15.74 (4.40) [10–33]
ADC-C	36.06 (7.64) [20–63]	35.04 (7.94) [20–63]	36.52 (7.46) [21–57]
ADC-Total score	68.01 (14.56) [41–121]	67.45 (15.82) [41–121]	68.27 (13.96) [41–117]

Note. The values are expressed as mean (standard deviation) [score range].

**Table 3 ijerph-20-01101-t003:** Descriptive analysis of variables.

Variables	Mean	SD	Min	Max
1. AQ-soc	1.98	1.74	0	8
2. AQ-switch	4.60	1.92	0	10
3. AQ-detail	5.50	2.22	0	10
4. AQ-comm	2.99	1.70	0	8
5. AQ-ima	2.55	1.61	0	8
6. EQ	47.18	10.39	17	73
7. SQ	31.47	11.75	4	73

Note. AQ-soc: AQ social skill; AQ-switch: AQ attention switching; AQ-detail: AQ attention-to-detail; AQ-comm: AQ communication; AQ-ima: AQ imagination. EQ: Empathy Quotient; SQ: Systemizing Quotient.

**Table 4 ijerph-20-01101-t004:** Intercorrelations between variables.

	1	2	3	4	5	6	7	8	9	10	11
1. ADC-A	-										
2. ADC-B	0.55 *** [0.49; 0.60]	-									
3. ADC-C	0.66 *** [0.61; 0.70]	0.60 *** [0.55; 0.64]	-								
4. ADC-Total	0.83 *** [0.80; 0.85]	0.80 *** [0.77; 0.82]	0.92 *** [0.90; 0.93]	-							
5. AQ-soc	0.14 ** [0.06; 0.21]	0.15 ** [0.07; 0.22]	0.26 *** [0.19; 0.32]	0.23 *** [0.16; 0.30]	-						
6. AQ-switch	0.20 *** [0.12; 0.27]	0.17 *** [0.09; 0.24]	0.27 *** [0.20; 0.33]	0.26 *** [0.19; 0.32]	0.36 *** [0.29; 0.42]	-					
7. AQ-detail	0.02 [−0.05; 0.09]	−0.08 [−0.15; 0]	−0.08 [−0.15; 0]	−0.06 [−0.13; 0.01]	0.03 [−0.04; 0.10]	0.03 [−0.04; 0.10]	-				
8. AQ-comm	0.33 *** [0.26; 0.39]	0.26 *** [0.19; 0.32]	0.34 *** [0.27; 0.40]	0.36 *** [0.29; 0.42]	0.45 *** [0.39; 0.50]	0.36 *** [0.29; 0.42]	0.04 [−0.03; 0.11]	-			
9. AQ-ima	0.13 ** [0.05; 0.20]	0.09 * [0.01; 0.16]	0.05 [−0.02; 0.12]	0.10 * [0.02; 0.17]	0.17 *** [0.09; 0.24]	0.14 ** [0.06; 0.21]	0.09 * [0.01; 0.16]	0.22 *** [0.14; 0.29]	-		
10. SQ	−0.10 * [−0.17; −0.02]	−0.17 *** [−0.24; −0.09]	−0.30 *** [−0.36; −0.23]	−0.24 *** [−0.30;−0.17]	−0.03 [−0.10; 0.04]	−0.09 * [−0.16; −0.01]	0.39 *** [0.32; 0.45]	−0.01 [−0.08; 0.06]	0.11 ** [0.03; 0.18]	-	
11. EQ	−0.22 *** [−0.29; −0.14]	−0.22 *** [−0.29; −0.14]	−0.25 *** [−0.31; −0.18]	−0.27 *** [−0.33; −0.20]	−0.28 *** [−0.34; −0.21]	−0.21 *** [−0.28; −0.13]	0.06 [−0.01; 0.13]	−0.44 *** [−0.49; −0.38]	−0.35 *** [−0.41; −0.28]	0.06 [−0.01; 0.13]	-

Note. AQ-soc: AQ social skill; AQ-switch: AQ attention switching; AQ-detail: AQ attention-to-detail; AQ-comm: AQ communication; AQ-ima: AQ imagination. *N* = 498; SQ: Systemizing Quotient; EQ: Empathy Quotient. * *p* < 0.05; ** *p* < 0.01; *** *p* < 0.001. [r 90% CI].

**Table 5 ijerph-20-01101-t005:** LASSO regression models.

Model/Factors	Goodness of Fit						Coefficients	
	Cohen’s f^2^	R^2^	R^2^ 90% CI	RMSE	MSE	Lambda	Coeff.	Beta Values
**1. ADC-A**	0.183	0.155	[0; 0.598]	0.92	0.86	0.03		
AQ-soc							0 *	0 *
AQ-switch							0.043	0.043
AQ-detail							0.011	0.011
AQ-comm							0.290	0.296
AQ-ima							0.030	0.030
EQ							−0.016	−0.016
SQ							−0.106	−0.104
**2. ADC-B**	0.162	0.140	[0; 0.583]	0.92	0.84	0.01		
AQ-soc							0.020	0.020
AQ-switch							0.057	0.057
AQ-detail							0 *	0 *
AQ-comm							0.171	0.175
AQ-ima							0.040	0.040
EQ							−0.100	−0.100
SQ							−0.175	−0.175
**3. ADC-C**	0.356	0.263	[0; 0.683]	0.87	0.76	0.02		
AQ-soc							0.080	0.081
AQ-switch							0.103	0.104
AQ-detail							0 *	0 *
AQ-comm							0.234	0.239
AQ-ima							0 *	0 *
EQ							−0.065	−0.065
SQ							−0.264	−0.261
**4. ADC-Total**	0.336	0.252	[0; 0.676]	0.87	0.77	0.01		
AQ-soc							0.041	0.041
AQ-switch							0.097	0.098
AQ-detail							0.012	0.012
AQ-comm							0.278	0.284
AQ-ima							0.018	0.018
EQ							−0.083	−0.083
SQ							−0.246	−0.242

Note. * Zero-correlation coefficients eliminated from the Model. AQ-soc: AQ social skill; AQ-switch: AQ attention switching; AQ-detail: AQ attention-to-detail; AQ-comm: AQ communication; AQ-ima: AQ imagination; EQ: Empathy Quotient; SQ: Systemizing Quotient.

## Data Availability

Raw data are available upon request from the first author.

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
