# Peer review of "The Relationships between Self-Reported Motor Functioning and Autistic Traits: The Italian Version of the Adult Developmental Coordination Disorders/Dyspraxia Checklist (ADC)"

_ijerph, 2023, doi:10.3390/ijerph20021101_

Round 1
Reviewer 1 Report
Thank you so much for giving me the opportunity to review this paper. I found is very informative since the scale presented in this paper could be helpful to identify dyspraxia in general population, but also in patients with neurodevelopment disorders. It is therefore important to validate the this scale in different languages. Overall the paper is very well written and also the quality of English is very good. I have only very small suggestions and after that I think that it could be published. I was wondering whether the authors could mention in the discussion more evidence for the use of this scale in patients with neurodevelopmental disorder? I was also wondering to how many languages it is currently translated?
Author Response
Comment 1. Thank you so much for giving me the opportunity to review this paper. I found is very informative since the scale presented in this paper could be helpful to identify dyspraxia in general population, but also in patients with neurodevelopment disorders. It is therefore important to validate the this scale in different languages. Overall the paper is very well written and also the quality of English is very good.
Response. We thank the Reviewer for these positive comments.
Comment 2. I have only very small suggestions and after that I think that it could be published. I was wondering whether the authors could mention in the discussion more evidence for the use of this scale in patients with neurodevelopmental disorder?
Response. We thank the Reviewer for having highlighted this aspect. In the revised Discussion section we provided a new paragraph on study limitations in which we discussed data on the use of the ADC with neurodevelopmental disorders, in particular with ADHD.
Comment 3. I was also wondering to how many languages it is currently translated?
Response. We have better specified in the Introduction section that the ADC is currently available in English, Hebrew and German.
Reviewer 2 Report
Dear authors, first of all, I would like to congratulate you for the work done. However, there are some minor changes that I would like you to correct before the final publication of the article.
First of all, I would opt for impersonal sentences throughout the manuscript (I would eliminate "we").
Introduction
What is DSM-V? Please explain.
It is indicated that most research on DCD is in children, stating the effects it has on them, but there is no mention of the effects on adults when it is later indicated that ADC is used in adults. Therefore, although it has been indicated that most research is on children and the effects it has on them, I would supplement the introduction with information on the effects of this disease in the adult population (The studies indicated in the "Aim 2" paragraph could be used to complete this section).
Why is this MT-16-19 Battery test used instead of the Handwriting Proficiency Screening Questionnaire used in previous research? Justification is needed for this and the psychometric properties of the MT-16-19 Battery test.
Method
There is a lack of information on the calculation of the sample size to know if it is representative of the study population.
There is a lack of information on the psychometric properties of the scales used.
Results
In the results section, it would be convenient to include in the tables the confidence intervals and the effect size of the tests that allow it. It would add statistical value to the article.
Discussion
At the end of the discussion, a paragraph summarizing the practical applications of this research, both in terms of the validation of the questionnaire and the differences found and the implications that this would have, is missing.
I would include the limitations of the study in a separate paragraph before the conclusions. In addition, they should better address the limitations of the study.
Conclusion
The conclusions are correct and are in accordance with the objectives.
Author Response
Comments to the Author
Comment 1. Dear authors, first of all, I would like to congratulate you for the work done. However, there are some minor changes that I would like you to correct before the final publication of the article.
Response. We thank the Reviewer for the comments that we allowed us further improve our manuscript.
Comment 2. First of all, I would opt for impersonal sentences throughout the manuscript (I would eliminate "we").
Response. Following the Reviewer’s suggestion, we modified the text throughout the manuscript (this change was not highlighted in yellow).
Comment 3. Introduction
Comment 3.1. What is DSM-V? Please explain.
Response. We thank the Reviewer for having highlighted the point. We explained the acronym.
Comment 3.2. It is indicated that most research on DCD is in children, stating the effects it has on them, but there is no mention of the effects on adults when it is later indicated that ADC is used in adults. Therefore, although it has been indicated that most research is on children and the effects it has on them, I would supplement the introduction with information on the effects of this disease in the adult population (The studies indicated in the "Aim 2" paragraph could be used to complete this section).
Response. We thank the Reviewer for the suggestion. We revised the Introduction section better describing the effects of DCD in adulthood.
Comment 3.3. Why is this MT-16-19 Battery test used instead of the Handwriting Proficiency Screening Questionnaire used in previous research? Justification is needed for this and the psychometric properties of the MT-16-19 Battery test.
Response. We decided to use the MT-16-19 Battery test because there isn’t an Italian version of the Handwriting Proficiency Screening Questionnaire with known psychometric properties. Furthermore, in the revised text we provided the psychometric properties of the test.
Comment 4. Method
Comment 4.1. There is a lack of information on the calculation of the sample size to know if it is representative of the study population.
Response. We thank the Reviewer for having highlighted the point. We have added these data in the revised text.
Comment 4.2. There is a lack of information on the psychometric properties of the scales used.
Response. We provided the missing data in the revised version of the manuscript.
Comment 5. Results. In the results section, it would be convenient to include in the tables the confidence intervals and the effect size of the tests that allow it. It would add statistical value to the article.
Response. We thank the Reviewer for the suggestion. In the revised text we provided the effect size (Cohen’s f2) and the confidence intervals for the four LASSO models (Tab. 5). As regards the Pearson’s correlation coefficients, we have now specified in the text that Cohen's guidelines (1988) have been considered for the purpose of interpreting the magnitude of a correlations. Therefore, we have only included the missing data (r’s confidence intervals) in both the revised text and the Table 4. In the same vein, we have added the confidence interval of the Cronbach αs (please, see Reliability section), as well as the score range for the ADC subscales and total score (Tab. 2). Finally, as far as Confirmatory factor analysis (CFA) is concerned, the model size effect is generally evaluated through the goodness-of-fit indices, including the Root Mean Square Error of Approximation index (RMSEA) and the Comparative Fit Index (CFI) (Shi et al., 2018; Kline, 2011; Cheung, 2002). Nevertheless, following the suggestion provided by the Reviewer, in the revised text we have also added the p values to the standardized saturations (Tab.1).
Comment 6. Discussion
Comment 6.1. At the end of the discussion, a paragraph summarizing the practical applications of this research, both in terms of the validation of the questionnaire and the differences found and the implications that this would have, is missing.
Response. Following the suggestion provided by the Reviewer in the successive point, in the revised Discussion section we provided a new paragraph on study limitations in which we took the occasion to discuss the practical implications of the present research, in particular focusing on the use of the Italian version of the ADC with neurodevelopmental disorders such as ADHD.
Comment 6.2. I would include the limitations of the study in a separate paragraph before the conclusions. In addition, they should better address the limitations of the study.
Response. We thank the Reviewer for having highlighted this aspect. We provided a new paragraph on study limitations in which we better discussed aspects of the study that needed to be considered in terms of limitations.
Comment 7. Conclusion. The conclusions are correct and are in accordance with the objectives.
Response. We thank the Reviewer for this positive comment.
Round 2
Reviewer 2 Report
The authors have successfully resolved all suggestions, so I recommend acceptance of the manuscript.